# Case Study of Multichannel Interaction in Healthcare Services

Ailton Moreira * , Júlio Duarte and Manuel Filipe Santos

ALGORITMI Research Centre, University of Minho, Azurém Campus, 4800-058 Guimarães, Portugal
* Correspondence: ailton.moreira@algoritmi.uminho.pt

**Abstract:** A multichannel interaction service is a practice whereby organizations communicate and interact with their existing customers and potential new customers through different channels. This article presents a brief case study of multichannel interaction in healthcare services, which studies the viability of continuous multichannel interaction for personalized healthcare services to enable health professionals to follow up and monitor patients in home-based care. Furthermore, this study aims to explore the possibility of the continuity and complementarity of the interactions across different communication channels with the patients. The data used for this study was gathered during the first wave of the COVID-19 pandemic. This study showed that despite this type of interaction being relatively new in healthcare services, it has considerable potential for improving the relationship between patients, health professionals, and care providers. Upon completion of the data analysis, several conclusions were drawn. One such conclusion was the ability to maintain continuity of interaction across multiple channels, as well as the synergy between the different channels of interaction available to patients and the impact this has on the way patients and health professionals interact. Additionally, it was determined that the complementarity of different interaction channels is crucial when implementing multichannel interaction services. Furthermore, the implementation of this solution resulted in improved communication between patients and health professionals. Also, it has decreased health professional's workload and reduced care providers costs regarding remote patient follow-up.

**Keywords:** multichannel interaction services; personalized healthcare services; multichannel services; continuity of healthcare services; conceptual model; COVID-19

## 1. Introduction

A multichannel interaction service is a practice that the retail industry has used to interact with its customers for a while. Following the literature, a multichannel interaction service is defined as using multiple channels in the communication and interaction process between customers and organizations. Furthermore, these organizations aim to take advantage of multichannel interaction to sell products and services to their customers or potential new customers. The evolution of technology has allowed retail organizations to target customers across multiple channels of interaction and retain their attention [1]. Through this type of interaction, retail organizations aim to provide their service to their customers across different channels of interaction [2].

Following the example of the retail industry, the healthcare industry has begun to incorporate this type of interaction in the services provided to the patient. This type of interaction in healthcare organizations is still in its early adoption [3].

In healthcare, multichannel interaction is still taking its initial steps, and it has the potential to change the way patients interact with health professionals. Thus, healthcare organizations aim to learn from the retail industry and apply the multichannel interaction strategy to the services that they provide to their patients [4].

This article aims to investigate the implementation of multichannel interaction in healthcare. Its goal is to study the viability of continuous interaction and personalized care

services across multiple interaction channels to enable patient follow-up and monitoring by health professionals. Furthermore, this study explores the possibility of continuity and complementarity in patient interactions across different communication channels.

This article presents the findings from the experiment carried out during the first wave of the COVID-19 pandemic in Portugal, in which a solution in the form of a multichannel interaction approach was applied. This study used data from patients who tested positive for the SARS-CoV-2 virus and were admitted for home-based care and were followed and monitored by health professionals. Upon the initial analysis of the data gathered during the first wave of the COVID-19 pandemic, it was possible to obtain fascinating insights regarding the impact of multichannel interaction on healthcare services. These insights are presented in detail in Sections 5 and 6.

Regarding the multichannel interaction in the healthcare services and the different characteristics of the available channels, a set of research questions (RQ) were established:

- RQ1—To what extent can the model provide continuity of service along multiple interaction channels?
- RQ2—To what degree does the model facilitate care process complementarity through the utilization of multiple interaction channels?
- RQ3—To what extent does the use of multichannel interaction improve the health professional's decision-making?

These research questions were intended to establish a guideline for this article regarding the analysis of the data collected and the contribution of multichannel interaction in healthcare services. The study was also designed to conduct an analysis regarding the average time the patient had to recover in home-based care, the average interaction between patients and health professionals via multiple channels, the impact of the utilization of multichannel services on patient relationships with health professionals, and finally, to identify which were the benefits that these patients received regarding the multichannel services. Likewise, a comparative analysis of the different channels used was carried out. Finally, the channels available to patients which had the most significant impact on their relationship with health professionals were explored.

This paper is divided into seven sections. The first section (Section 1) introduces the study and outlines the main research questions. The second section (Section 2) provides a background for multichannel interaction in general and for healthcare services in particular. The third section (Section 3) presents the research methods used in the study, including data extraction and an overview of the data. The fourth section (Section 4) shows the COVID-19 workflow implemented at CHUP and used in the study. The fifth section (Section 5) presents the main results of the data analysis. The sixth section (Section 6) includes a discussion of the findings. Finally, the seventh and final section (Section 7) presents the study's conclusions and highlights the research's main contributions.

## 2. Background

Multichannel interaction services can be defined as strategies that take advantage of multiple channels to communicate with customers at some point in their customer journey. The capacity to use available technology to ensure a specific target audience is presented with information or the ability to react to information across multiple channels [5,6]. Customers expect personalized, relevant communications to grab their attention despite their busy schedules. They are also more likely to react to an organization's message if it is delivered through their preferred media (interaction channel). In practice, this involves sending the appropriate message, at the right time, via the right channel [7]. Multichannel interaction services are more of an operational approach that allows customers to complete transactions using different channels [7]. An illustration of multichannel interaction (Figure 1) shows that the customer has access to various interaction channels that he/she can use to interact with an organization.

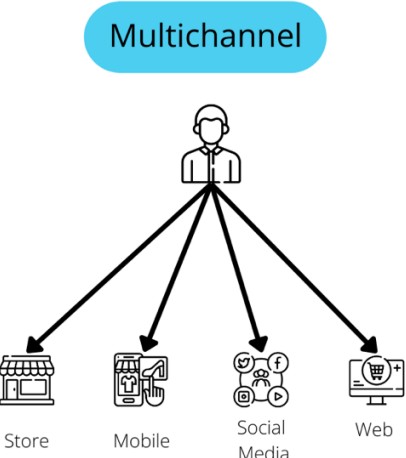

**Figure 1.** Illustration of multichannel interaction in a generic use case.

Concerning health care, in an equivalent way to clients, patients also tend to respond more actively to stimuli with the most appropriate content and time.

*2.1. Benefits of Multichannel Interaction*

Multichannel interaction services provide numerous benefits to both organizations and customers. Organizations can take advantage of this trend to increase sales and business growth. Using multiple channels to interact with customers enables a much wider reach, making organizational business potentially visible to new customer groupings. Some benefits of multichannel interaction that have been identified in the literature are [6,8–10]:

- Customer relationship and acquisition—organizations can efficiently target individual customers through their preferred channels at the right moment.
- Increase customer satisfaction and loyalty—organizations aim to provide clear, targeted, and consistent customer communication via any channel.
- Cross-sell/up-sell—organizations aim to transform regular transactional documents, such as invoices or statements, into relevant marketing tools containing personalized offers to specific customers.
- Reduce outsourcing cost—organizations can bring document creation in-house instead of delegating it to outsource companies.
- Improve organization brand—organizations that offer their customers a more fluid and convenient interaction journey across multiple channels will result in better customer experience, and this ends up generating higher levels of customer satisfaction. Satisfied customers positively impact the organization's brand image because they are far more likely to recommend their products or services to their friends and families.
- Improve operational efficiency—organizations can merge data and streamline business processes to create, produce, deliver, and track customer communications across multiple channels.
- Ensure brand consistency—organizations aim to control and manage all the critical and sensitive data from the organization and its customers.

Multichannel interaction has many more benefits for the organization, and these benefits can increase business growth. The organization needs to be able to communicate and interact across all channels to maintain a close relationship with customers and keep them satisfied. However, even more important than keeping this relationship, it is crucial for organizations to know how, when, and in which channel they should interact with their customers because customers appreciate receiving this interaction at the right time, in the channel that they prefer [5,6,9,10].

### 2.2. Challenges of Multichannel Interaction

Although multichannel interaction services provide many benefits to organizations and customers, they present some challenges that organizations must overcome to provide a better and more secure user experience. The significant challenges identified in the literature regarding multichannel interaction are [5–7,11–14]:

- Data integration—the organization must decide which data they should integrate across multiple channels (e.g., purchase data only or should search data be included as well).
- Customer behavior—organizations should understand how customers choose channels and their impact on their overall purchasing patterns.
- Channel valuation—the organizations retrieve data and understand the customer decision-making process, and they can evaluate channel performance.
- Resource allocation—an organization's channel policy is manifested in its resource allocation across different channels.
- Channel strategies—the organizations' most challenging task is coordinating the objectives, design, and deployment of channels to create synergies.

The challenges identified above raise many other questions about customer data privacy and an organization's data security policies. Organizations must identify these issues to enable them to better understand and prepare their IT infrastructure and their strategy for the multichannel environment [7].

### 2.3. Multichannel Interaction in Healthcare Services

Multichannel interaction in the healthcare service (MIHS) is an approach that is based on interactions between patients (through their preferred channels of interaction) and health professionals, as well as the technical IT infrastructure required to create the multichannel interaction environment [3,4]. Multichannel interaction services in healthcare are in their early adoption. Multichannel interaction aims to significantly impact the healthcare services provided to patients by enabling newer and better ways of interaction between patients and healthcare professionals [15]. A multichannel interaction conceptual model was proposed for patients' healthcare service across different channels [4].

The proposed model is more decentralized and focused on the patient (patient-centered) and not a centralized model focused on the healthcare entity. According to the multichannel interaction model applied in the retail industry, the proposed model has four key areas: people (patients and health professionals), channels, services, and care providers. They represent the critical dimensions of multichannel interaction in healthcare. They extend to the three basic concepts of healthcare service architecture (services, people, and providers) with the central concept of channel management (channels). In a multichannel healthcare environment, each patient or group of patients has different needs (diseases), preferences for services, and channel usage [16,17].

### 2.4. Multichannel Interaction Conceptual Model

Figure 2 introduces the conceptual model of multichannel interaction for health services: MIHS [4]. The model has three main tiers: patient tier, coordination tier, and care-provider tier.

To coordinate the interaction between patients and health professionals and all clinical records of patients, the platform for interaction, integration, and interoperability of hospital systems was used as the base support system. The Agency for Integration, Diffusion, and Archive of Medical Information (AIDA) was proposed as an interoperability platform designed to address the problem of the lack of an integration tier to handle information from multiple systems with a focus on interoperability, confidentiality, integrity, and data availability. AIDA is a platform based on multi-agent technology systems to make Hospital Information Systems (HIS) interoperable [18]. AIDA has intelligent agent systems that ensure interoperability between heterogeneous information systems [19]. In a multichannel environment, healthcare organizations take advantage of the Hospital 4.0 environment [4] and new trends to offer better and improved patient services through multiple interaction

channels. This development has positively impacted healthcare organizations' services and patient relationships. Furthermore, patients also value using their preferred channels to interact with healthcare organizations when they need them the most [20,21].

The core of the proposed conceptual model is the coordination tier. It ensures the continuity, integrity, interoperability, security, and privacy of all interactions between the actors involved and the clinical data of patients exchanged in these interactions.

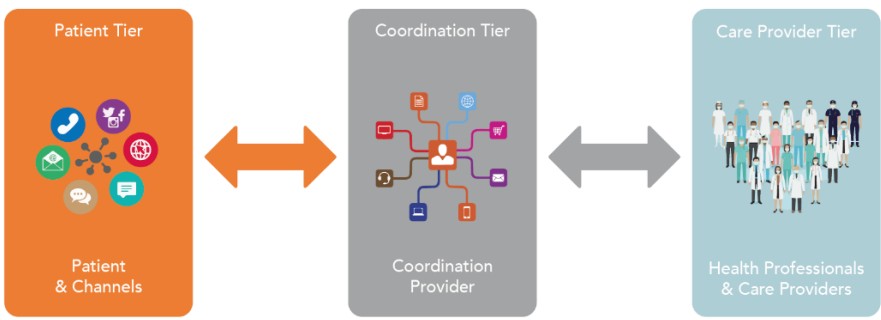

**Figure 2.** Conceptual model of multichannel interaction.

## 3. Materials and Methods

### 3.1. Data Extraction Strategy

The data used for this analysis were extracted from *Centro Hospitalar Universitário do Porto* (CHUP) and from interactions between patients and health professionals during the period that the patients were under medical surveillance in home-based care. They were extracted between March and July 2020, a period in which a high number of patients were in home-based care during the first wave of COVID-19.

### 3.2. Data Collected

After retrieving the data, the initial findings obtained from it are presented in Table 1.

**Table 1.** Patient interactions.

| Type of Interaction | Total of Interaction |
|---|---|
| CHUP Monit (web application) | 11,236 (interactions from patients in home-based care) |
| Phone contact | 565 (interactions from patients in home-based care) |

The data for this study were collected from patients who had tested positive for COVID-19 and received home-based care during the analysis period. A total of 638 patients were included in the study, and their interactions with healthcare professionals were recorded and analyzed. The data showed that these patients had 11,801 interactions with healthcare professionals during the analysis. Most of these interactions, approximately 95%, took place through the CHUP Monit web application, while the remaining 5% occurred via phone. The data presented in Table 1 provides insights into the prevalence and methods of interaction between COVID-positive patients receiving home-based care and healthcare professionals. Upon further analysis of the collected data, several characteristics were identified. These included the average duration of medical surveillance for these patients, the average patients' age, and the average daily interactions between patients and healthcare professionals, among others. A detailed analysis of these findings is presented in Section 5 (Results).

### 3.3. Methods

The research method selected for this study is a case study; a qualitative research method that allows for the in-depth examination of a specific phenomenon within its context. The case-study method [22–24] is particularly well suited for exploring the relationship between the phenomenon of multichannel interaction in the context of healthcare

services and the people (patients and health professionals) involved. In addition to utilizing a case study approach, this study also utilized a quantitative research technique, which involves collecting and analyzing data that is numerical and statistical. It is used to understand patterns and relationships in data and to make predictions about future outcomes. It includes descriptive statistics (summarizing and describing data), inferential statistics (predicting outcomes based on a sample), and multivariable analysis (analyzing data with multiple variables). Furthermore, it allows for informed decision-making based on statistical evidence.

## 4. Case Study: CHUP COVID-19 Workflow

This article aims to present the findings regarding the impact of using the multichannel interactions model implemented at Centro Hospitalar Universitário do Porto (CHUP). The data used in this study were collected in Portugal during the first wave of the COVID-19 pandemic. The experimental approach comprised two interaction channels integrated with the hospital's health record system. Although the MIHS is still at an early stage of adaptation, there are already some conceptual models for its implementation. The present study is supported by the previously presented conceptual model for multichannel interaction in healthcare services [4].

Users increasingly depend on new interaction channels to communicate with organizations [25], which was reinforced by the COVID-19 health crisis. Based on this principle, health organizations have also started to adhere to this paradigm by making services available to their patients through multiple interaction channels. The conceptual model [4] was combined with new applications designed and integrated with third-party applications in healthcare systems. With the combination and integration of these applications, it was possible to identify all the main tiers of the conceptual model [4].

This article presents a specific case study for implementing the multichannel interaction of patients admitted for home-based care during the first wave of the pandemic in Portugal. It is based on the multichannel model presented earlier and implemented at CHUP. It focuses only on the COVID-19 workflow that is detailed in Figure 3.

Given the high number of patients infected with the COVID-19 virus, the hospital units could not accept all these patients. In order not to overload these units, patients with COVID-19 who had no symptoms (asymptomatic patients) were referred to home-based care. For these patients, a set of channels (CHUP Monit and a phone number) were made available for interaction with health professionals to receive medical monitoring and follow-up, as well as report the daily evolution of their health condition regarding their symptoms of COVID-19.

As mentioned previously, this article aimed to carry out a case study of the utilization of the different channels of interaction during the period of home-based care from the moment the patients contacted the hospital until they were discharged, to study the impact that MIHS had on health organizations as well as on patients. Based on the principle of MIHS, Figure 3 simplifies the workflow that was followed to filter patients with COVID-19 who were admitted to home-based care.

According to the COVID-19 workflow presented in Figure 3, it is evident that the interaction begins with the initial contact with the CHUP (either through telephone referral or urgent care). Then it continues with assistance from the COVID-19 services in the urgency service, triage, and referral for hospitalization or home-based care for patients with low severity or absence of symptoms. Patients hospitalized for home-based care continue to receive medical follow-ups. Once these patients no longer have symptoms for a specified period, test negative for the SARS-CoV-2 virus, or meet regulatory health restrictions, they are discharged.

Patients in home-based care who experience worsening symptoms, as determined by data collected through continuous monitoring, are readmitted for hospitalization, where they can receive face-to-face monitoring by healthcare professionals.

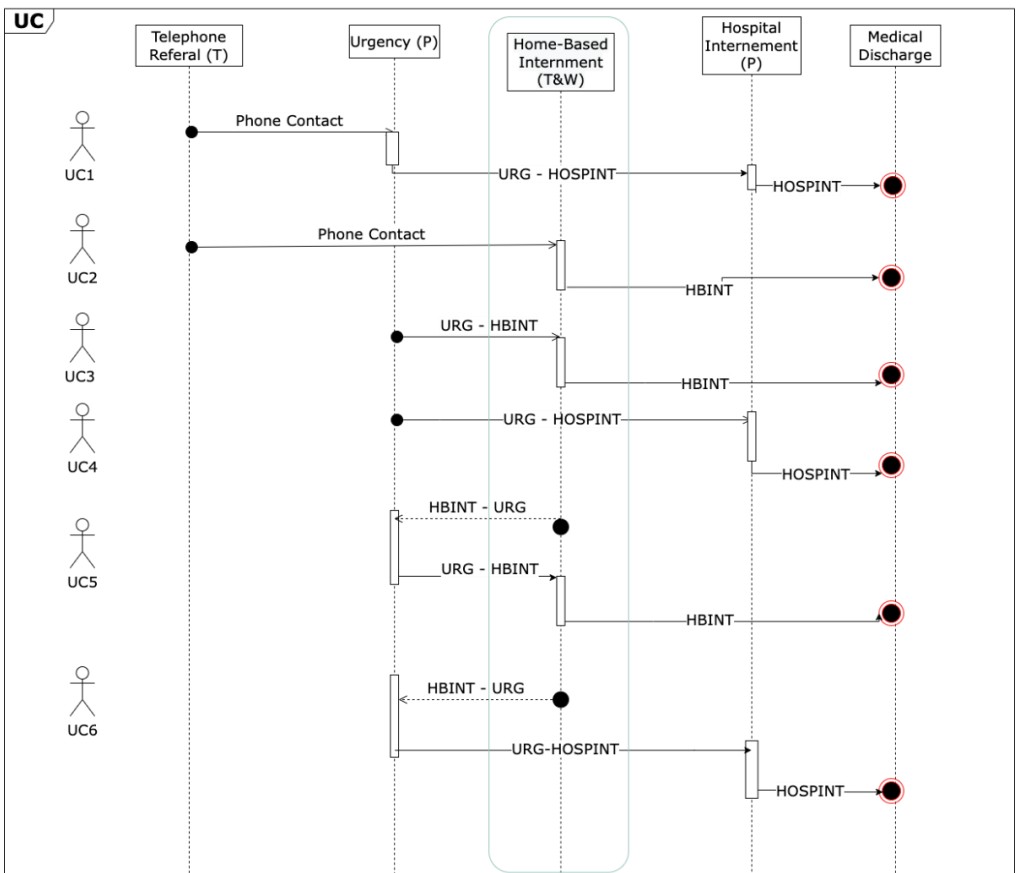

**Figure 3.** COVID-19 workflow.

As shown in Figure 3, use cases 1 and 2 depict the workflow initiated through telephone contact, with the patient being admitted to hospitalization in the first case and home-based care in the second case. In use cases 3 and 4, the workflow starts with an in-person visit to the CHUP emergency services. Through the COVID-19 testing and triage service, the patient is admitted for home-based care if they do not present severe symptoms of COVID-19 (use case 3) or is admitted to the hospital if they present severe symptoms (use case 4). Finally, use cases 5 and 6 portray patients who are in home-based care, but experience worsening symptoms and end up going to the CHUP emergency services. In use case 5, the patient is screened and, as the symptoms are not severe, is directed towards home-based care again, while in use case 6, the patient is admitted to the hospital due to more severe symptoms caused by COVID-19.

Two channels were made available to patients to communicate with health professionals:

1.  CHUP Monit—was a web application developed specifically for patients admitted for home-based care to interact with health professionals and record the evolution of symptoms caused by the COVID-19 virus during the period they were under medical surveillance. Below are some images of CHUP Monit (web application) designed and made available to patients under home-based care surveillance to interact with health professionals. The following image labels are presented in Portuguese because it was a solution designed and implemented in Portugal.

Access was granted via the mobile phone number associated with the patient, who would therefore receive a pin via SMS for authentication. After successful authentication, the patient was redirected to the home page of CHUP Monit with authentication (patient area). On the home page of CHUP Monit, the patient had a list of all the previous interactions with health professionals and all the previous information that they had submitted (see Figure 4).

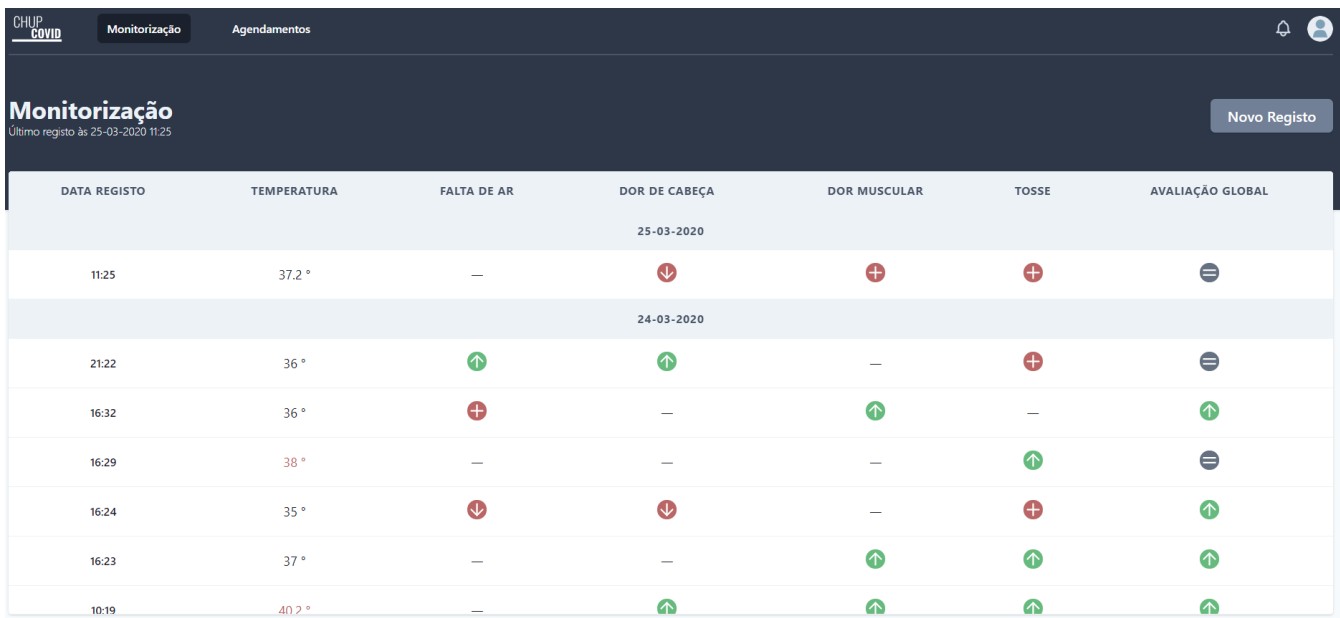

**Figure 4.** CHUP Monit home page (history data of previous interactions).

To submit a new interaction (a form with symptoms), the patient could use the button "Novo Registo" (new register) to start a new interaction. Then a window was opened with a form that the patient must fill in with the symptoms they had at that moment (see Figure 5). On the auto-surveillance form, the patient had to insert the information about their body temperature, symptoms (typical symptoms of SARS-CoV-2, shortness of breath, headache, muscle pain, cough, etc.), a global assessment of their health condition, and medication in the last 24 h.

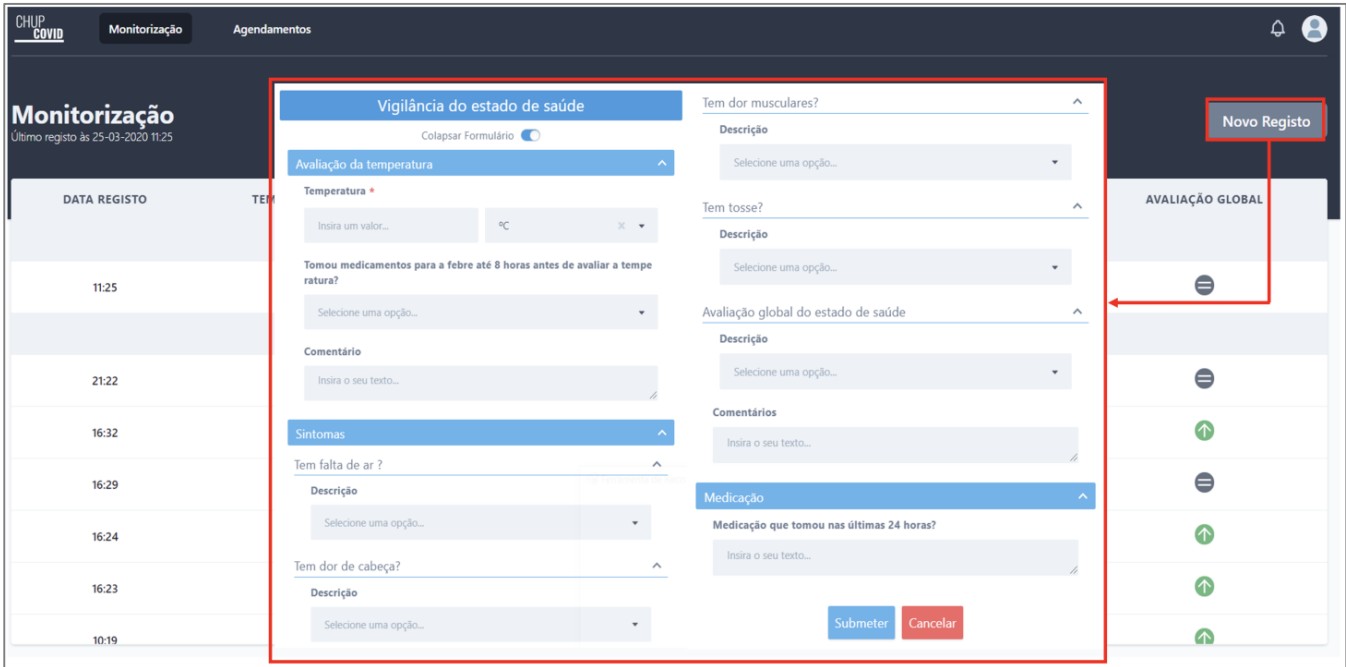

**Figure 5.** CHUP Monit new form submission to interact with health professionals.

Phone contact—patients admitted to home-based care had a phone number to contact health professionals directly as an alternative channel of interaction with health professionals while they were under medical surveillance.

These two channels allowed health professionals to interact with, monitor, and follow up patients regarding the evolution of the symptoms caused by the COVID-19 virus.

## 5. Results

As mentioned previously, the quantitative method of analysis was applied. Alongside this, a set of statistical analyses were also performed.

When patients were in home-based care, they had to interact with health professionals to receive medical care follow-up regarding the evolution of symptoms of SARS-CoV-2. There was considerable adherence in interactions between patients and health professionals based on the interval of days that they had been under medical surveillance (see Figure 6).

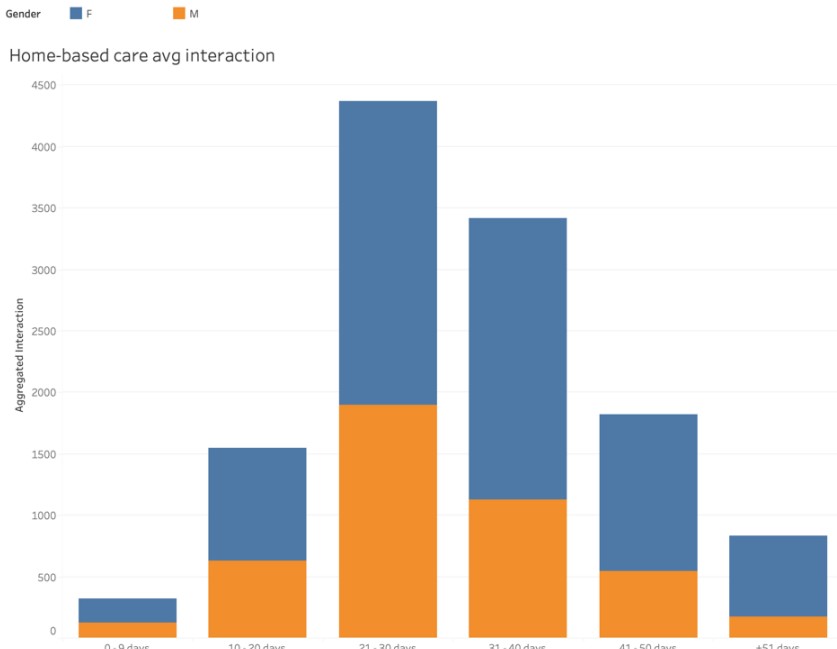

**Figure 6.** Average days of patient home internment.

Furthermore, as shown in Figure 6, the patients with 21–30 days of medical surveillance were the ones who had more interaction with health professionals. From that group, the female patients interacted with caregivers the most. Following that interval, were the patients with 31–40 days of medical follow-up in terms of interaction with health professionals. Again, female patients had more interaction than male patients. After 31–40 days, there was a decline in interaction with health professionals, mainly because not many patients had persistent symptoms for an extended period. In general, female patients had more interaction with health professionals.

When patients were admitted to home-based care, they received instructions on interacting with health professionals and received medical care follow-up. Initially, only one channel (phone contact) was available for patients to interact with health professionals. That channel alone was not enough, and it was a synchronous interaction channel that required a health professional's availability to register the patient's clinical data. Instead of reducing caregivers' workload, using only these channels increased the workload of those professionals. Another interaction channel was introduced shortly after due to the caregiver-workload increase (CHUP Monit). The new channel introduced was an asynchronous interaction channel, which did not require direct contact with caregivers and could reduce the workload of those professionals.

Upon in-depth analysis of Figure 7, it could be stated that, in general, patients had a strong adherence to and usage of the digital channel of interaction that was introduced instead of the manual channel to interact with health professionals. The CHUP Monit (web application) was the preferred interaction channel for patients to interact with health professionals, given its portability, ease of use, and low-waiting time to interact with health professionals. Furthermore, it could be concluded that in terms of continuous interactions, the CHUP Monit (web application) had a significant impact on the model proposed. Its daily use was much more significant than phone calls (another channel made available to interact with health professionals).

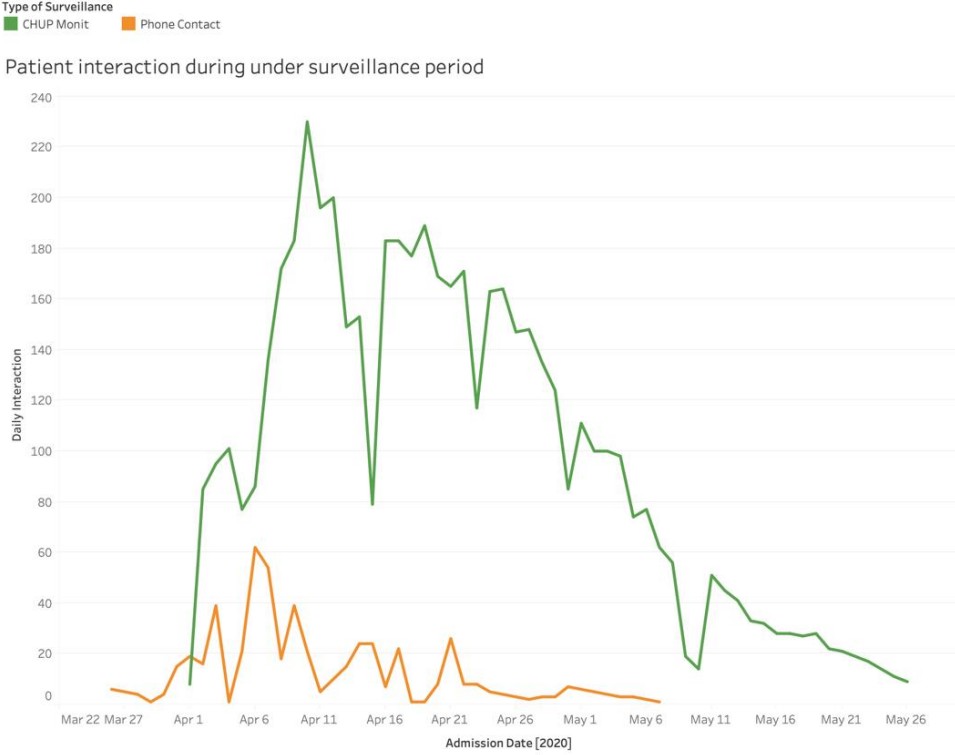

**Figure 7.** Average daily interaction with CHUP Monit and phone contact.

The period in which the CHUP Monit (web application) had the most daily use coincided with the period in which there was the most significant number of interactions with health professionals. The CHUP Monit (web application) was the primary driver in these interactions, reaching almost 240 daily interactions with health professionals. Health professionals used phone calls to contact patients that had not used the CHUP Monit (web application) for any reason and to give patients medical discharge. In these cases, phone calls were used to reach the patients that had not submitted their COVID-19 symptoms.

An analysis of the data gathered by age group is presented in Figure 8, and it represents the average patient interaction in home-based care based on their group. This graph enabled the correlation of patients' age groups with the average interaction in each group. Patients between 21 and 80 years old had a significant number of interactions, but patients in the groups of 31–40 years and 51–60 years old had the most interaction with caregivers. Furthermore, female patients had more consistent interaction with caregivers (in all age groups).

Figure 9 shows the interaction distribution by channels. With some analysis techniques, it was found that the CHUP Monit (web application) alone represented approximately 71.11% of interactions between patients and health professionals. Likewise, it was noted that about 21.41% of interactions between health professionals took place using both interaction channels (web application and phone contact). Finally, it was noted that phone contact represented only 7.48% of interactions between patients and health professionals.

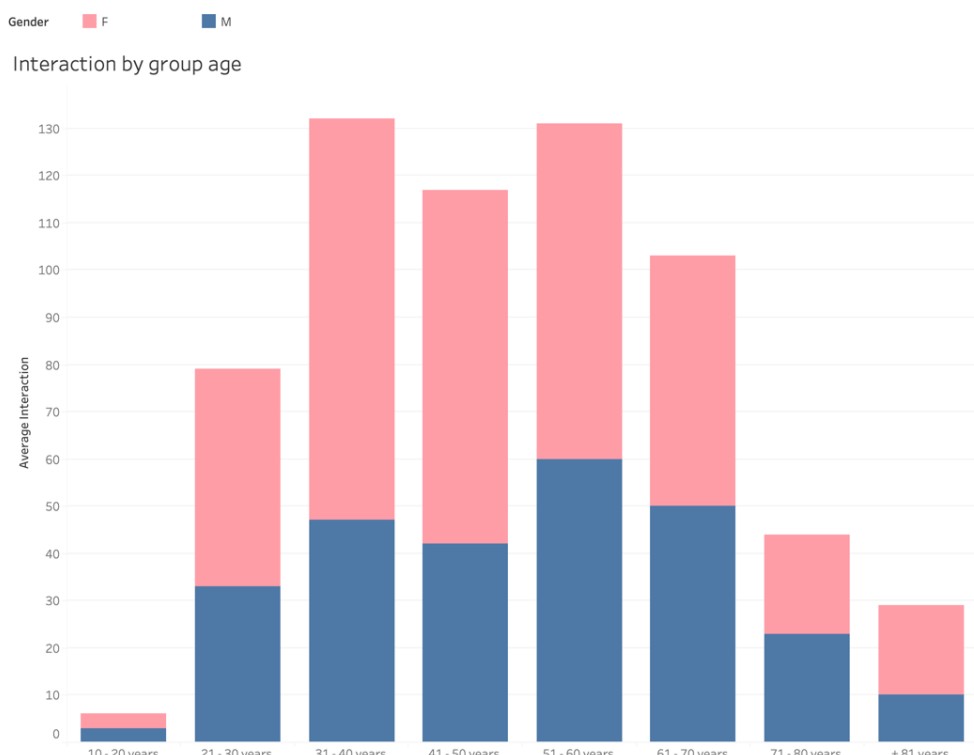

**Figure 8.** Multichannel interaction by age group.

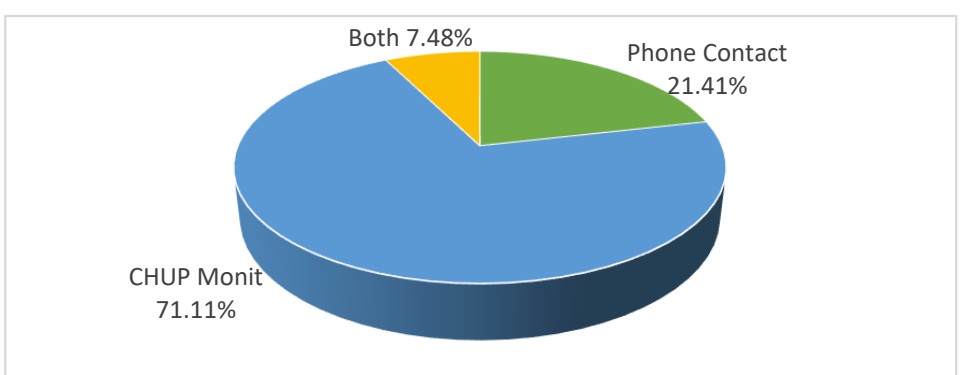

**Figure 9.** Distribution of interaction by channels.

According to health professionals, they often used phone contact (synchronous channel) to interact with patients who, for any reason, did not make the daily registration of the evolution of symptoms. Furthermore, they used this channel to give patients medical discharge if they lacked symptoms of COVID-19 and had fulfilled the criteria for medical discharge. Often, health professionals used phone contact to interact with patients to gain more insight into their evolution and health conditions regarding COVID-19 symptoms.

Figure 10 shows the relationship between the days of home-based care and the interactions between patients and health professionals. This chart highlights that a considerable number of patients stayed, on average, less than 50 days in home-based care. It was the period in which there was the highest number of interactions between health professionals and patients. With the increase in days of home-based care, there was a decrease in the number of patients in home-based care and a consequent decrease in the number of interactions between health professionals and patients.

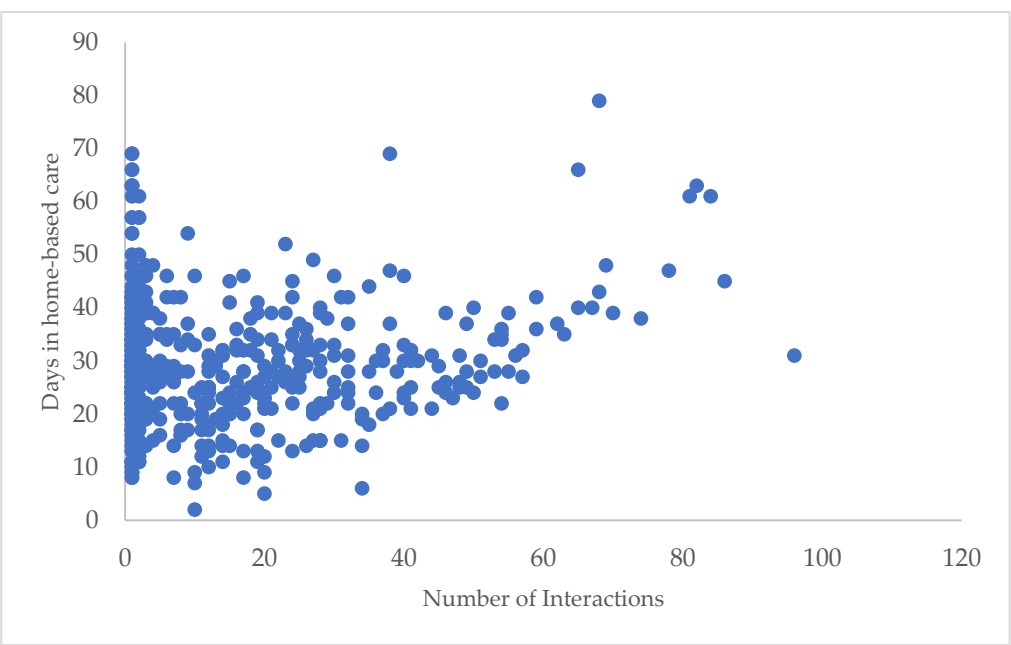

**Figure 10.** Daily interaction vs. days in home-based care.

## 6. Discussion

This section presents a brief discussion of the findings gathered from the data and the corroboration of the research questions outlined in this article. Nevertheless, a Spearman correlation was applied to the data under analysis before answering these research questions. A statistical method was applied for the data analysis to test the hypothesis. The correlation methods aim to identify whether there is any correlation between two or more variables. In this case, it was intended to determine if there was any correlation between the continuity and complementarity of different interaction channels available to patients. The correlation between the two variables is measured through the value of $\rho$, which can have a robust correlation ($\rho$ = 0.9 to 1), strong ($\rho$ = 0.7 to 0.9), moderate ($\rho$ = 0.5 to 0.7), weak ($\rho$ = 0.3 to 0.5) and very weak ($\rho$ = 0 to 0.3) and can be positive or negative.

Spearman Correlation—hypothesis test

**H0.** *The channels of interaction used by patients are not associated, i.e., independent.*

    **vs.**

**H1.** *The channels of interaction used by patients are associated, i.e., they are dependent.*

The hypothesis test was performed on the patient's channels (phone contact and CHUP Monit) to establish the relationship between these two variables. The hypothesis test was then elaborated from these two variables, whose $\rho$ value was obtained at 0.000 (see Table 2).

**Table 2.** Spearman correlation.

| **Spearman Correlation** | |
| --- | --- |
| Coefficient | −0.7630 |
| N | 2411 |
| *t* statistic | 57.952 |
| DF | 2409 |
| *p* value | 0.000 |

In the statistical analysis of the gathered results, the ρ value was less than 0.05, which means the H0 was rejected at the 5% significance level because there was statistical evidence that the two variables under analysis were associated, i.e., they were dependent. The H1 (hypothesis alternative) was accepted. From the analysis of the data presented, the following conclusions were found:

RQ1—To what extent can the model provide continuity of service among multiple interaction channels?

With the results gathered from the Spearman correlation, the chart in Figure 11 shows that these channels were indeed associated. Furthermore, in the beginning, patients only used phone contact to interact with health professionals. However, shortly after the CHUP Monit was introduced, they continued their interaction using the new channel available to interact with health professionals. Despite the patient's preference for using the web application (CHUP Monit) to interact with health professionals at any moment, they could change to another available channel (phone contact) and continue the interaction without losing the information previously registered on the web application [26].

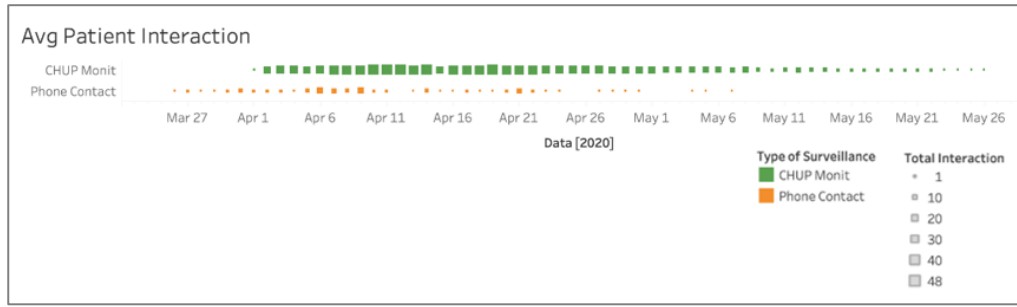

**Figure 11.** Continuity of interaction using different channels.

The continuity of interaction was secure through the coordination tier, and it was possible to store all the interactions that patients had carried out in both channels. The implementation of the coordination tier ensured all interoperability between the different channels and HIS used to record and present the patient clinical information in a single touchpoint for the health professionals [8,12].

RQ2—To what degree does the model facilitate care process complementarity through the utilization of multiple interaction channels?

From the chart in Figure 9, patients who used more than one channel to interact with health professionals were selected (7.48%). Based on this selection, a new chart was drawn to ascertain whether there was a care process complementarity in using the different interaction channels. Alongside this, a Spearman correlation was performed regarding the patients who used both interaction channels.

Figure 12 shows the period in which both interaction channels were being used simultaneously, proving that, to a certain extent, there was indeed a care process complementarity in the use of the multiple channels of interactions. This was corroborated in a case where a patient presented an unfavorable evolution of the symptoms. The patient was then contacted (through telephone contact) by the health professional to better understand the evolution of the symptoms that they had, according to the physicians. Furthermore, from Figure 9, it can be concluded that the use of the two channels (web application and phone contact) was not proportional, and the chart based on the selected group of patients presented in Figure 12 was indeed a care process complementarity of the utilization of multiple channels of interaction. Furthermore, with the Spearman correlation results at 5% significance, the alternative hypothesis (H1) was accepted, which means there was indeed a complementarity between these two channels of interaction.

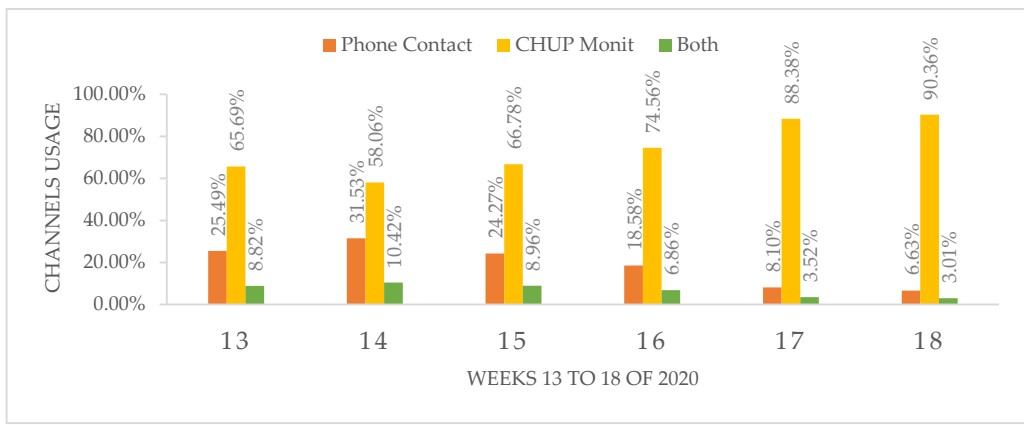

**Figure 12.** Channel utilization complementarity.

From the analysis of the results obtained, it can be concluded that it was possible to maintain the continuity of the services in multiple channels of interaction [RQ1]. There was a care process complementarity in the use of the channels available to patients, as in some cases, phone contact was often used to complement the use of the web application [RQ2]. The utilization of the phone contact channel in these situations allowed health professionals to have more direct insight regarding the patient's health condition than they would have obtained from the interaction through the web application [8,12,14,27–29]. They maintained synchronous communication, which was more effective and efficient in providing information about the evolution of patients' symptoms and making decisions about medical discharge depending on the evolution of the symptoms.

Although most patients preferred the web application, health professionals could only discharge their patients by telephone. Given this fact, it was concluded that there was a complementarity in using the different interaction channels used by patients in interactions with health professionals.

RQ3—To what extent does the use of multichannel interaction improve decision-making by health professionals?

With the different types of analysis carried out in this paper, some interesting conclusions were drawn regarding the utilization of multiple interaction channels and how it relates to health professional activities. Due to the availability of patient clinical data at the right moment, the health professionals had better timing in their decision-making because they had access to a patient's clinical record about the evolution of the symptoms of COVID-19 in real-time [30]. These data were presented to them in a helpful format that could help them in their decision-making process [3,31].

Given that patients adhered to the channels available to interact with health professionals, it made the decision-making of the health professionals who accompanied these patients easier, as they always had access to the patient's clinical data regarding the evolution of COVID-19 symptoms recorded by patients. This timely and practically real-time access facilitated the health professionals' analysis of these data and the consequent decision-making arising from these data. It could culminate in the patient's medical discharge if he/she did not present worsening symptoms and had already fulfilled the period of isolation, or the admission of the patient to the hospital unit for physical follow-up if the patient had an unfavorable evolution of symptoms [32–34]. The CHUP Monit (web application) played an essential role. The data retrieved from the web application and presented to health professionals was already processed and transformed (Figure 10). It was the channel that the patients used the most.

Exciting conclusions were drawn from the data gathered about the patients who had more than 25 days of home-based care internment and showed an unfavorable evolution in their symptoms of SARS-CoV-2. These patients who had some worsening COVID-19

symptoms such as shortness of breath and very high fever and reported these symptoms to health professionals were admitted again to the hospital units where they continued to receive physical medical follow-ups by health professionals. It is worth noting that at the time of the recommendation for home-based care, patients were still under medical surveillance. In the case of worsening symptoms, they were admitted to the hospital and continued to be followed physically by health professionals. This fact led to a high number of days of hospitalization (days under medical surveillance), that is, the interval between the first time these patients contacted hospital services with symptoms of COVID-19 until they effectively had a medical discharge.

Although the number of days in the hospital may have been high, the number of days with symptoms was usually lower. Even in cases where patients no longer had symptoms, they were still under medical surveillance until they were discharged due to restrictions imposed by health regulatory authorities.

Multichannel interaction can bring enormous advantages to the organization that adopts it. In this case, implementing the proposed model for multichannel interaction during the first wave of COVID-19 at CHUP showed some advantages for the healthcare organization. The main advantage was the reduction in effort in human resources management, costs, and logistics for the monitoring and follow-up of these patients. The utilization of digital channels reduced the health professional's need to contact patients and interact with them [35,36].

*SWOT Analysis*

A SWOT analysis, as shown in Table 3, allows for the systematic evaluation of the potentialities and weaknesses of multichannel interaction in healthcare services. This is an analysis that contributes to the validation of new paradigms, complements the data analysis conducted, and highlights the benefits of multichannel interaction in healthcare services [25,37,38].

**Table 3.** SWOT analysis of the model of MIHS.

| Strength | Weakness |
|---|---|
| A model with a considerable impact on health organization<br>Increase in services available through multiple channels<br>Extended channels of interaction with patients<br>Increased patient satisfaction and loyalty | Lousy implementation of the multichannel model<br>Poor patient targeting across multiple channels<br>Health professionals not well prepared for MIHS with patients<br>Additional costs associated with implementation and maintaining multichannel interaction |
| **Opportunities** | **Threats** |
| More services available through multichannel of interaction<br>Improve the quality of services offered to patients and business<br>Healthcare organization cost reduction<br>Provide personalized healthcare services to patients | Laws regarding the patient's data security and privacy<br>Government regulations regarding the MIHS<br>Model subject to approval by healthcare authorities |

Finally, with all the data analysis completed, the proposed multichannel interaction model was met with high levels of acceptance from both patients and health professionals. Through the application of this model, this study was able to address the challenge of examining the feasibility of providing continuous healthcare services to patients across multiple channels of interaction, as well as enhancing the quality-of-care services delivered to patients. The quantitative analysis of the data collected demonstrates that implementing the multichannel interaction model in healthcare services facilitated the ongoing monitoring and support of patients receiving medical care follow-up in home-based care. Furthermore, this study opened the door to additional potential applications of this interaction model in healthcare services for patients.

## 7. Conclusions

One of the significant challenges that healthcare institutions face today is integrating and managing patient-generated data obtained through different channels of interaction. To address this challenge, it was essential to develop a mechanism to facilitate the inclusion and accessibility of patient-generated data within the Healthcare Information Systems (HIS). The conceptual model designed for this purpose enabled the integration and management of HIS. Within the model, the coordination tier played a critical role in systems integration and interoperability operations, as well as ensuring the seamless continuity of services across different communication channels.

Given the characteristics in the coordination layer implemented, the patient could begin interacting with health professionals on a specific channel. The patient could change to another channel and continue interacting with the health professionals without losing information regarding previous interactions. The coordination layer implemented makes possible the mapping and persistence of data in the different interaction channels, which enables the continuity of the interaction of the different channels between patients and health professionals. With the use of the model proposed for healthcare services, health professionals could interact with patients interned at home and offer continuous and personalized medical service to those patients using different interaction channels.

Based on the implementation of multichannel interaction at CHUP, the study conducted in this article aimed to present the contributions of the multichannel interaction model in healthcare services regarding the continuity of services across different channels of interaction. This study was conducted within the context of applying this interaction model during the first COVID-19 wave at the CHUP. The model presented was applied to patients in home-based care. A set of interaction channels were made available to these patients, which they could use to interact with health professionals. The analysis of the data gathered from the application of this interaction model allowed us to conclude that the model had a significant impact on the interaction between patients and health professionals, and the health organization itself.

The multichannel interaction model implemented had a significant adherence regarding the channels available to patients to interact with health professionals. According to the data analysis, it was apparent that the coordination of different channels of interaction and the integration of different hospital systems were essential to provide a continuous service to patients across multiple channels.

The analysis in this article allowed us to perceive and validate by applying the conceptual model in the practical context of MIHS. From the study, it can be concluded that multichannel interaction is a viable practice in healthcare services, bringing new opportunities that enable interaction between patients and health professionals. Its practice dramatically facilitates the communication process between both parties, which often did not happen previously, as there was no way of multichannel interaction combined with the continuity of healthcare services provided to patients.

This study focused on analyzing the results obtained during the COVID-19 pandemic regarding the feasibility, practicability, and continuity of MIHS and the consequent validation of the proposed interaction model in the previous studies [4].

This approach is still in its very early phase of healthcare adoption so further studies will be needed concerning MIHS adoption. From the various conclusions drawn from the present analysis, the most significant contribution that most caught our attention was the ability to provide continuity of interaction between multiple channels and the synergy between the different channels of interaction available to patients, as well as the impact that this synergy had on the way patients and health professionals interacted. Complementarity between different channels is a factor that must be considered when implementing multichannel interaction services.

The results of this study are in line with other studies [3,4,25,38] in suggesting that the proposed interaction model has enormous potential in the healthcare services offered to patients by health professionals, as well as for the health organization and their business

model. The bottom line is that multichannel interaction in healthcare services can bring innumerable advantages to care providers, health professionals, patients, and society. This study has proven that despite its substantial initial effort and implementation in the initial phase, it will pay off all the initial efforts in the long term. Finally, the main highlights identified with the implementation of MIHS include a reduction in workload for caregivers, cost reduction with telephone carriers, timely access to patient clinical data for decision-making purposes, improved patient satisfaction with their participation in the follow-up process, and continuity of the patient follow-up across multiple channels of interaction. Additionally, the advantage of this study lies in developing and implementing a multichannel interaction model for healthcare services, focusing on analyzing this model's feasibility, practicability, and continuity during the COVID-19 pandemic. This study presents the first application and analysis of this interaction model in a practical context in a Portuguese healthcare institution, providing valuable insights into the impact of multichannel interaction on patient–healthcare professional interactions and the healthcare institution as a whole. Through the analysis of data gathered from the implementation of this model, it was concluded that multichannel interaction is a viable practice in healthcare services that brings new opportunities for interaction and facilitates communication between patients and health professionals. The ability to maintain continuity of interaction across multiple channels and the synergy between these channels were found to have a significant impact on the way patients and healthcare professionals interact. This study highlights the importance of coordinating and integrating different channels of interaction and hospital systems to provide continuous service to patients. Furthermore, this study highlights the complementarity and continuity of interaction between multiple channels of interaction in healthcare services. Further studies on adopting multichannel interaction in healthcare services will be necessary to fully understand this approach's potential and limitations.

**Author Contributions:** Conceptualisation, A.M. and M.F.S.; methodology, A.M. and M.F.S.; software, A.M.; validation, J.D.; formal analysis A.M. and J.D.; investigation, A.M., J.D. and M.F.S.; resources, A.M. and J.D.; data curation, A.M. and J.D.; writing—original draft preparation, A.M.; writing—review and editing, A.M., J.D. and M.F.S.; visualization, J.D.; supervision, M.F.S.; project administration, M.F.S.; funding acquisition, M.F.S. All authors have read and agreed to the published version of the manuscript.

**Funding:** This work has been supported by—Fundação para a Ciência e Tecnologia, within the R&D Units Project: UIDB/00319/2020. Ailton Moreira was support by the grant 2020.10342.BD.

**Institutional Review Board Statement:** The study was conducted under the Declaration of Helsinki and approved by the Ethics Committee of Centro Hospitalar Universitário do Porto for studies involving anonymized patients' data regarding their interaction with health professionals.

**Informed Consent Statement:** Not applicable (all data used in this study was completely anonymized).

**Data Availability Statement:** The dataset analyzed during the current study is not publicly available due to the Administrative Council of Centro Hospitalar Universitário do Porto's authorization for research purposes only and not for publication but is available from the corresponding author on reasonable request.

**Acknowledgments:** This work has been supported by—Fundação para a Ciência e Tecnologia, within the R&D Units Project: UIDB/00319/2020. Ailton Moreira was support by the grant 2020.10342.BD.

**Conflicts of Interest:** The authors declare no conflict of interest.

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
