# Peer review of "Case Study of Multichannel Interaction in Healthcare Services"

_information, doi:10.3390/info14010037_

Round 1

Reviewer 1 Report (Previous Reviewer 2)

This article focuses on an example of omnichannel engagement in healthcare services, which examines the viability of continuous omnichannel engagement for personalized healthcare services so that healthcare professionals can observe and manage patients at home. The relevance of the study is justified by the fact that the service of multi-channel interaction is a practice in which organizations communicate and interact with their existing and potential new customers through different channels. This study aims to explore the possibility of continuity and complementarity of interactions with patients through various communication channels. The data used for this study was collected during the first wave of the COVID-19 pandemic. The study found that while this type of interaction is relatively new in health services, it has significant potential to improve relationships between patients, health professionals, and healthcare providers.

Despite the satisfactory quality of the article, some shortcomings need to be corrected.

  1. It is recommended to expand the abstract with concrete results obtained within the research.
  2. The data used for the study should be described in more detail.
  3. The methods section should be expanded by highlighting the approaches proposed by the authors.
  4. The workflow presented in Figure 3 should be described in detail in the text.
  5. Figure 4 does not show any novel information or results, and can be deleted.
  6. The scientific novelty of the research should be highlighted.

In summarizing my comments I recommend that the manuscript is accepted after minor revision. 

Author Response

Dear Reviewer,

Thank you so much for taking the time to review our paper, "Case study of Multichannel Interaction in Healthcare Services." We greatly appreciate the thoroughness of your review and the valuable comments and suggestions you provided.

Your feedback has been extremely helpful in improving the quality of the manuscript and we have incorporated many of your recommendations into the revised version. We hope that the changes made address your concerns and meet the standards for publication.

Once again, we thank you for your time and effort in reviewing our work. Your insights have been invaluable, and we are grateful for the opportunity to benefit from your expertise.

Sincerely,

The authors

Reviewer 2 Report (Previous Reviewer 1)

The authors have sufficiently addressed my comments from a previous review. I have some minor concerns:

In the SWOT analysis, the weaknesses part is not really comprehensible. Some of the entries are unclear. E.g., is the first line really a weakness?

Further, on the negative side, one important issue seems to be missing: Patient-generated data are difficult to fit into the HIS. Often, some workaround is necessary. I think that this subject is a major challenge.

Several of the Figures are not clearly readable. Please also revisit the captions of the figures.

Figure 1: caption: -> "Illustration of Multichannel Interaction in a generic use case"

Figure 4: "Login" -> "Login window" or "Login page" ... or similar. Please also add a frame around, so that it becomes clear that the Covid virus is a part of the login window.

Figure 5: the content is not readable. Would it be possible to make this clearer? Possibly also add some legend. Please also revisit the caption.

Figure 6: the content is not readable at all. Would it be possible to make this clearer? Please also revisit the caption.

Table 2: Do you really need as many digits behind the ","? Three or four digits should suffice.

There is a variety of quite long sentences that appear incomprehensible. Please revise these sentences. Examples: Lines 520-525: this paragraph is incomprehensible and appears vague.

Lines 574ff: please rewrite.

Lines 503-507: very long sentence ... please rewrite

Line 65: some -> an

Something is wrong with the indentation in the References section.

Author Response

Dear Reviewer,

Thank you so much for taking the time to review our paper, "Case study of Multichannel Interaction in Healthcare Services." We greatly appreciate the thoroughness of your review and the valuable comments and suggestions you provided.

Your feedback has been extremely helpful in improving the quality of the manuscript and we have incorporated many of your recommendations into the revised version. We hope that the changes made address your concerns and meet the standards for publication.

Once again, we thank you for your time and effort in reviewing our work. Your insights have been invaluable, and we are grateful for the opportunity to benefit from your expertise.

Sincerely,

The authors

This manuscript is a resubmission of an earlier submission. The following is a list of the peer review reports and author responses from that submission.

Round 1

Reviewer 1 Report

The manuscript presents a study of feedback to the hospital during the first wave of COVID-19 using two modalities: by phone, app-based, or both modalities in a mixture. The properties of both modalities are vaguely described. Further, modalities such as video-consulting, automatic transfer of sensor data, communication with AI-based chat-engines for non-urgent communication are not considered in this work. Thus, the term multi-modality might not be appropriate to use, as only two modalities are compared.

The term multi-modality is vaguely defined. Due to language issues, I did not understand the definition of the term. Also the term "omni-channel interaction" remains unclear, including which role it plays in the current manuscript. The concept of multi-modality using the three layers is rather high-level. It remains unclear, which role this model plays in your research, and whether there are alternative architectures that would have an impact on your results. Instead of "layer", I would recommend using the term "tier", as the parts refer to physically separated entities. (I recommend a web search "tier vs layer").

There are language issues that make large parts of the manuscript incomprehensible, and the reader might misunderstand the content.

Most of the figures (specifically Figs. 2, 3, 5) include text that is too tiny. Such text would be not readable in normal print. Please use font sizes that are at least twice as large for these small fonts.

The SWOT analysis in the Conclusion Section should come much earlier in the manuscript. Note that parts of the SWOT analysis seem to be implementation dependent, rather than generic regarding specific implementations.

The research questions are named H1..H3. However, later in the manuscript, there are hypotheses that use H0 and H1. This is confusing. Further, note that it remains unclear how the research questions are answered.

The experiment setup is not clearly described. Although the number of patients is mentioned, it is unclear how many patients participated, and how many patients used which modality at which point in time. I assume that all the ethical forms are appropriately approved (as this is more strict in a medical context).

There might be several biases that might have an impact on the results. These are not further discussed. For example, it seems that the app started later than the phone-regime. Which impact could this have on the results. Further, from the age-distribution, it seems that the first two age groups are differently defined than the others (10 yrs each). The first (0-8) also would suggest that the parents would answer, while for the second (9-19) this would be rather unclear.

Further, it is unclear how patients that are admitted to the hospital after some stay at home are treated in your study. There are some comments about this, but the impact on the results is unclear.

It seems that there is no difference between the two modalities. It is not made clear which possible mechanism would suggest differences between these modalities regarding the duration in a non-acute situation. A further bias might be the age profile, as older people might have a different interaction pattern compared to younger people, and also older people might have a different impact regarding COVID-19. This is not further evaluated in the study.

Unfortunately, I don't understand Figure 7 and the discussion around this subject. Possibly, two diagrams would be clearer? Please explain this in a better way. What values are on the x and y axis?

Figure 8 is unclear.

I am not quite sure what Figure 4 shows.

Regarding the results for the health professional, the app has different properties from the phone modality. The app uses asynchronous communication, and the access the appropriate health records is done automatically. For the phone modality, one could also automatically access the appropriate health record, e.g., from the phone number. This could be implemented in the back-end (Care Provider Layer); however, this seems not to be implemented. Therefore, the mentioned differences are highly implementation dependent. That is, you compare the implementation of the app with the implementation of the phone modality. This is, however, not further outlined in your manuscript.

Author Response

Dear review,

Thank you for taking the time to review our paper and provide your valuable insights and suggestions. We truly appreciate your careful consideration of our work and the thorough commentary that you have provided. Your feedback was invaluable as we worked to improve the paper and make any necessary revisions. We look forward to the opportunity to address your comments and incorporate your suggestions in the revised version of the paper. Thank you again for your time and expertise.

With best regards,

The authors

Reviewer 2 Report

This article focuses on omnichannel engagement in health services, which explores the viability of continuous omnichannel engagement for personalized healthcare services to enable patient follow-up and monitoring by healthcare professionals. The study's relevance is justified by the fact that the service of multi-channel interaction is a practice in which organizations contact and interact with their customers and potential new customers through several channels of interaction. The study explores the possibility of continuity and complementarity of interactions between different communication channels with patients. The data used for this study was collected during the first wave of the COVID-19 pandemic. The study showed that although this type of interaction is relatively new in healthcare, it has great potential to improve relationships between patients, healthcare professionals, and healthcare providers.

Despite the satisfactory quality of the article, some shortcomings need to be corrected.

  1. Moving the research questions and aims to the Introduction part is recommended.
  2. Expanding the current research analysis with research on medical information systems is recommended. E.g., http://ceur-ws.org/Vol-2753/paper19.pdf
  3. The source where the authors get the data should be defined and described.
  4. The proposed web app should be described in more detail.
  5. It is recommended to discuss the use of the web app. How does the use of the app affect compared to the lack of such an application?
  6. It is recommended to shorten the Conclusions and to move some paragraphs to the Results and Discussion sections, e.g., SWOT analysis.
  7. The scientific and practical novelty of the study should be highlighted.

In summarizing my comments, I recommend that the manuscript is accepted after major revision, including a detailed description of the information technology proposed within the research. 

Author Response

(The authors gave the same response as above.)

Reviewer 3 Report

Multichannel interaction is an important and promising way of delivering healthcare services. However, the contribution of the current study on this research filed is not clearly described.

1.     In Section 2, H1, H2 and H3 represent “research questions”. However, in Section 4, they represent hypotheses where appears H0 which is not mentioned previously. The hypotheses are simply mentioned without any theoretical basis.

2.     The study is lack of a sound theoretical foundation. The literature is weak. References are old. Almost all the references are published about 5 years ago.

3.     The Title of the manuscript says “case study”, however, it does not mention case study as a research method in the main body of the manuscript.

4.     Data analysis should be conducted in a proper way. For example, Table 1 shows only two types of interactions, i.e, phone contact and web app. However, Figure 6 shows that there are three types of interactions.

5.     There are inconsistencies in the results. For example, it says “From the analysis of the new graph presented in Figure 9, it can be concluded that initially there was only one channel of interaction available to patients to interact with health professionals, but shortly afterward another channel of interaction was introduced.” (p.12) Figure 9 starts on Mar 27. However, as shown on Figure 4, the data available starts earlier than this date and that there are more web app interactions than telephone contact before Apr 1 which is not consist with Figure 9.

6.     Discussions need to be revisited. Most of the discussion has been done without references. There is little discussion pertinent to previous studies.

7.     Revisit your theoretical implication. Did not find any theory and you have concluded your theoretical implications.

8.     The manuscript needs to be restructured, especially Section 2.

9.     The figures are hard to understand. There are no axis labels. Captions of some figures are not useful summarization of the content.

10.  Table 1 is also hard to understand. The first line of a table usually contains labels of each column. The sum of web app and phone contact does not equal to patient interaction.

Author Response

(The authors gave the same response as above.)
